# A New Strategy for Somatotype Assessment Using Bioimpedance Analysis: Stratification According to Sex

**DOI:** 10.3390/jfmk7040086

**Published:** 2022-10-14

**Authors:** Alexander Bertuccioli, Davide Sisti, Stefano Amatori, Fabrizio Perroni, Marco B. L. Rocchi, Piero Benelli, Athos Trecroci, Francesco Di Pierro, Tindaro Bongiovanni, Roberto Cannataro

**Affiliations:** 1Department of Biomolecular Sciences, University of Urbino Carlo Bo, 61029 Urbino, Italy; 2Department of Biomedical Sciences for Health, University of Milan, 20126 Milan, Italy; 3Digestive Endoscopy Unit and Gastroenterology, Fondazione Poliambulanza, 25124 Brescia, Italy; 4Scientific & Research Department, Velleja Research, 20125 Milan, Italy; 5Galascreen Laboratories, University of Calabria, 87036 Rende, Italy; 6Research Division, Dynamical Business & Science Society, DBSS International SAS, Bogotá 110311, Colombia

**Keywords:** body composition, linear models, somatotypes, bioimpedance

## Abstract

Body composition assessment is a relevant element in the biomedical field, in research and daily practice in the medical and nutritional fields, and in the management of athletes. This paper aimed to operate in an Italian sample investigating the possibility of predicting the somatotype from bioimpedance analysis and comparing the predicted results with those obtained from anthropometric measurements. This observational study was conducted with retrospective data collected from 2827 subjects. The somatotype of each subject was calculated both with the Heath–Carter method and by a multiple regression model based on bioimpedance and anthropometric parameters. Somatotypes (endomorph, mesomorph, and ectomorph) were predicted with a high goodness of fit (R^2^ adjusted > 0.80). Two different somatocharts were obtained from anthropometric measures and bioimpedance parameters and subsequentially compared. Bland–Altman plots showed acceptable accuracy. This study could be a first step in developing a new approach that allows the detection of a subject’s somatotype via bioimpedance analysis, stratified according to sex, with a time-saving and more standardized procedure. It would allow, for example, during the COVID-19 pandemic, to minimize operator–patient contact in having measurements.

## 1. Introduction

Body composition assessment is considered a key factor for the evaluation of the general health status of humans, and its measurement have increasingly been considered valuable in clinical practice [1]. For example, in pharmacology, it allows studying the distribution space of a drug within an organism; in nephrology and cardiology, to identify early dehydration or hyper-hydration conditions secondary to pathology; and in epidemiology, to study the connection between morbidity and body composition. Body composition is integrated into some areas with morphological analysis to more clearly define a condition or the results of certain interventions. In sports, it allows us to verify changes in muscle mass by monitoring the conditions of the athlete after specific training programs or specific workloads and in nutrition, it allows us to define the nutritional and energy needs of a patient by subsequently monitoring the positive or negative effects of a diet. Anthropometry is the most used outpatient methodology for assessing body composition in its various meanings, from estimating functionality to assessing the state of health and its related prognostic indices [2]. The term “somatotype” was coined for the first time in 1940, and it is defined as the description and quantification of the present morphological conformation and composition of the human body. It is expressed in a three-number rating, representing endomorphy, mesomorphy, and ectomorphy components, respectively, always in the same order [3,4]. It comprises of three models that, in their development, ideally outlined the prevalence of one of the three embryonic sheets: endoderm, mesoderm, and ectoderm [5]. Sheldon et al. [5] further defined the three fundamental somatotypes, identifying in the endomorph a rounded and soft structure; in the mesomorph, a structure with developed muscle and robust skeleton; and in the ectomorph, a structure with a fragile and delicate build, together with a long-line structure [4]. However, this analysis was conducted only through photographic observation. The limitations of this analysis were overcome by the Heath–Carter method [4], which is based on anthropometric measurements such as mass, stature, skinfolds, breadths, and girths.

Researchers and professionals must acquire considerable skill in making the measurements, demonstrating consistency in the superimposition of the values detected in multiple measurement sites of skinfolds in the same subject and making multiple measurements on the same day, on consecutive days, or even weeks away [6,7]. A limit is represented by the difficulty of measurement in subjects who are too thin or too fat and the impossibility of detecting the visceral adipose tissue, whose expansion is not correlated with that of the subcutaneous adipose tissue. Furthermore, in subjects with a high degree of obesity, the maximum opening of the skinfold caliper may not be enough. In these circumstances, the measurement is not performed, or—by convention—a value equal to 45 mm is assigned. Furthermore, causes of unreliable measurements can be: (1) choice of a measurement site is not well located; (2) incorrect application of the skinfold caliper (as a result of which it is either too inclined, too superficial, or too deep); (3) a plica is raised with the fingers; and (4) early or delayed reading time. The Heath–Carter method is one of the most used techniques in evaluating the somatotype; it requires the acquisition of different anthropometric measures common to different techniques used for the assessment of body composition [3]. However, the above-described drawbacks limit the availability of the Heath–Carter method for large-scale studies.

Significant correlations of some somatotypes with hormonal responses to stress in young soccer players suggest that it could be an element of interest in the selection process and training planning [8]. The first step to validating a method valid for specific populations of athletes is to study its applicability in the general population.

A technique developed in the 1990s for studying body composition is the vector analysis of the bioelectrical impedance (BIVA), which also allows for evaluating the variations that occur due to hydration and nutrition conditions [9]. BIVA is based on the electrical properties of biological tissues. Electricity flows easily along highly conductive body tissues if a weak alternating electric current is injected into the body [10]. It does not passthrough cells; instead, the current flows through the extracellular fluid [11]. However, the cell membrane capacitor charges and discharges the current at a high frequency. Thus, current flows through the cell membranes and tissue fluids [12]. The volume of water determines the width of the passage through which electricity flows, represented by impedance. Bioimpedance—or biological impedance—is defined as the ability of biological tissue to impede electric current [13]. BIVA measures resistance (R) and reactance (Xc) values. R represents the opposition to the flow of an alternating current through intra- and extracellular ionic solutions, whereas Xc represents the capacitive component of tissue interfaces, cell membranes, and organelles [14,15]. Due to the non-invasiveness, relatively low cost, and portability of the BIVA, many studies on body composition and clinical condition evaluations have been carried out using this technique [11].

A limitation of the bioimpedance analysis is the assumption that the human body is an isotropic (equivalent electrical properties regardless of the measurement direction) conductor with homogeneous length and cross-sectional area, and that it complies with the precautions that theoretically could compromise full standardization (repeated measurements at the same time of the day, no large meals 2–4 h prior to testing, no coffee or alcohol consumption >8 h prior to testing, consumption of liquids limited to 1% of body mass within 2 h prior to testing, no heavy physical activity >8 h prior to testing). Further attention should be paid to diuretic medications, metallic jewelry, preparation of the skin surface, quality and positioning of the electrodes, and thermoneutral conditions [16,17]. Previous studies have shown that it is possible to predict, with a good margin of accuracy, a result that can be superimposed on the evaluation of the somatotype according to Heath–Carter through a BIVA evaluation [18,19,20,21]. Furthermore, correlating with other parameters such as muscle quality, performance, and injury indices [22,23] suggests how probable the compositional and morphological study of the organism carried out with different techniques can represent two different points of view, which, when suitably integrated, are able to provide the same information with a good level of overlap.

The present work aimed to investigate the possibility of predicting the somatotype with a BIVA vector analysis (a technique conventionally used in the assessment of body composition) in the Italian population, comparing the predicted results with those calculated using standard anthropometric measurements. This methodology would allow us to standardize the measurement technique and reduce the possibility of human errors related to multiple measures and the time needed to evaluate the somatotype, defining new potential applicability for BIVA analysis. The differences in body composition between women and men, even if not directly considered in the acquisition of the anthropometric parameters according to Heath–Carter and of the bioimpedance parameters, could constitute a relevant element in the evaluation of the results, for which a stratification by sex becomes important in the study.

## 2. Materials and Methods

### 2.1. Participants

This study was conducted on a dataset of 2827 patients living in the Marche, Emilia Romagna, and Lombardia Italian regions, from 2019–2021. It was a retrospective study, since a nutritionist previously collected the data used for the analyses during his professional practice. All data were anonymous; participants signed a personal data treatment informed consent, and the study was conducted according to the Declaration of Helsinki; the Ethics Committee approved the work protocol of the University of Milan (2019-Approval code number: 1052019). Data from male and female subjects were used; inclusion criteria were: age between 18 and 65, good general health, absence of chronic diseases in the active phase, absence of acute diseases, and absence of limb prostheses or joint prostheses. The following constituted exclusion criteria: sporting activity at a professional or competitive level, intake of diuretic, antihypertensive drugs capable of acting on the dynamics of hydration and on the intestines, intake of nutraceuticals capable of acting on the dynamics of hydration, intake of drugs, and presence of outcomes of skin lesions capable of interfering with a correct anthropometric and bioimpedance analysis.

### 2.2. Anthropometric Assessment

Anthropometric measurements such as stature and mass, subscapular, triceps, iliac crest and medial calf skinfolds, joint breadths of the biepicondylar humerus and biepicondylar femur, and arm (flexed and tensed) and calf girths were taken according to the standard methods [3,4,24] and used for the calculation of the somatotype according to the Heath–Carter method [3,4,25]. All the anthropometric measurements were performed by the same operator, a physician nutritionist who had been working at a clinical level for over 15 years with more than 10,000 patients assessed When considering the general, sportive, and clinical fields, the physician nutritionist performed at least 25 assessments/week. Each parameter was measured three times (median value used), and random checks were carried out to constantly monitor the technical measurement error (TEM), verifying that it remained within the stability parameters of Carter’s operating manual. (TEM for skinfolds should be approximately 5%, for widths and circumferences 1%, and for stature about 0.5%) [3]. Mass (to the nearest 50 g) and stature (to the nearest 0.1 cm) were measured with a SECA 700 Eye Level Beam Mechanical Column Scale (SECA North America–Medical Measuring Systems and Scales, 13601 Benson Avenue, Chino, CA, USA). Skinfolds were taken with a caliper Metrica 70200. Circumferences were measured with Metrica 38910 metric tape and amplitudes joints with a Metrica 10460 small sliding clamp (Metrica Via Grandi, 18, 20097 San Donato Milanese, Italy). Girths and joint breadths were taken to the nearest 0.1 cm, while skinfold was taken to the nearest 0.2 mm [24]. All the instruments used that provided for the possibility of calibration were calibrated before each measurement session according to the instructions provided by the manufacturers.

### 2.3. Bioimpedance Vector Analysis

The impedance values, evaluated in its R and Xc components, were obtained using a bioimpedance analyzer (BIA 101 Anniversary Sport Edition, Akern, Florence, Italy) in combination with the Bodygram Plus^®^ software (Akern, Florence, Italy) and Biatrodes electrodes, strictly following the manufacturer’s protocol. The sample enrolled in this study did not follow any specific instructions (e.g., diet, hydration, physical activity) before the BIA measurements. The subjects laid in a supine position on a non-conductive surface for at least 2 min before the measurement, allowing a homogeneous distribution of body fluids. Subjects were positioned with a leg opening of 45° to the body midline, and upper limbs were positioned 30° away from the trunk, verifying the absence of contact with any metal object and the absence of contact between the upper limbs and the trunk and between the lower limbs. After cleansing the skin with alcohol, four electrodes (Biatrodes, Akern Srl, Florence, Italy) were applied, with 5 cm between them; two electrodes were place on the back of the right hand, in correspondence with the radioulnar joint and the metacarpophalangeal joint of the third finger; the other pair of electrodes were placed on the back of the foot, one on the tibiotarsal joint and the other in correspondence of the metatarsophalangeal joint of the third toe [17,26,27]. The data obtained were processed using the Bodygram Plus software by analyzing the direct values and obtaining the relative derived estimates. The room temperature was between 22 and 26 °C for all measurements.

### 2.4. Statistical Analyses

Descriptive statistics were reported as the mean (SD) for quantitative variables. A multivariate analysis of variance was used to test differences between sexes for all demographic and anthropometrics variables considered. A multiple regression model was used: the single dependent continuous variables were “endomorphy”, “mesomorphy”, and “ectomorphy;” the measures were obtained using anthropometric evaluation. The independent variables were all the measures, conventional and vector, derived from bioimpedance analysis: Rz (resistance), Xc (reactance), FFM (free fat mass), TBW (total body water), ECW (extracellular water), BCM (body cellular mass), FM (fat mass), PA (phase angle), NA/K (Na/K ratio), FM% (fat mass percentage), FFM% (free fat mass percentage), TBW% (total body water percentage), ECW% (extracellular water percentage), ICW% (intracellular water percentage), MM (muscular mass), MM% (muscular mass percentage), Mbasal (basal metabolism), BCMI (body cellular mass index), hydration (score of the subject’s hydration status), nutrition (score of the subject’s state of nutrition), SMI (skeletal muscle mass index), SMM (skeletal muscle mass), ASMM (appendicular skeletal muscle mass), FMI (fat mass index), and FFMI (free fat mass index). Furthermore, regression models considered age, stature, mass, and BMI. Regressions were conducted separately for males and females. To avoid multicollinearity between the variables, VIF (variance inflation factor) was calculated; some variables were eliminated until the VIF were all less than 10. The multiple regression model used for the high number of variables analysed was the forward stepwise regression. The final result is several models and their summary statistics. Stop criterion and goodness of fitting index was R^2^ adjusted (R^2^_adj_), which quantifies what percentage of the variability in the dependent variable is accounted for by all of the independent variables together; dependent variables were endomorphy, mesomorphy, and ectomorphy values obtained by skinfold measures, as described above.

Endomorphy_regr_, mesomorphy_regr_, and ectomorphy_regr_ morphotype values, predicted by multiple regressions, were utilized for obtaining the coordinate of a Reuleaux triangle (known as somatochart), with the following linear combinations:X_regr_ = ectomorphy_regr_ − endomorphy_regr_
Y_regr_ = 2 × mesomorphy_regr_ − (ectomorphy_regr_ + endomorphy_regr_)

Therefore, X_regr_ and Y_regr_ were linearly dependent on the ectomorphy, mesomorphy, and endomorphy parameters obtained from the regression; the mean of residual error was different from the expected value, equal to zero (E_(e)_ = 0). This means that a linear transformation (X_regr_corr_ = ax + bx X_regr_; Y_regr_corr_ = ay + by Y_regr_) can be used in order to meet the regression assumption again. In order to compare agreement between methods, the Bland–Altman plot analysis was used to evaluate a bias between the mean differences and to estimate an agreement interval, within which 95% of the differences of the proposed regression-based method, compared to the anthropometric measures’ method, fell. Lin’s concordance correlation coefficients were also calculated. All the elaborations were performed using Microsoft Excel and SPSS Statistics (version 22.0; IBM, Chicago, IL, USA).

## 3. Results

A total of 2827 subjects (42.1% males and 57.9% females) took part in this study. Males and females were significantly different in age and anthropometric characteristics, apart from body mass index (*p* = 0.117). Participants’ characteristics are reported in Table 1.

### 3.1. Multicollinearity

Before using multiple regressions, the VIF values were checked. Next, the following variables, which showed multicollinearity, were removed: mass, BCM, FM, FM%, FFM%, ECW%, ICW%, ideal body mass, nutrition, SMM, and FMI. Once the variables were removed, multicollinearity was no longer evident, as all the VIF values relating to the remaining variables were less than the value of 10. The final prediction models are reported in Table 2. 

### 3.2. Endomorph Somatotype

Multiple linear regression was calculated to predict endomorph somatotype values. For males, a significant regression equation was found (F_(3,1279)_ = 899.8, *p* < 0.0001), with an R^2^ adjusted of 0.678, considering, at the end of the forward selection, only three variables. The variable associated with endomorph values with a positive coefficient was the BMI; the variables negatively associated were stature and TBW%.

The male predicted endomorphy value measurement was:Endomorphy value = 10.44 − 0.0297 × Stature − 0.0683 × TWB% + 0.150 × BMI

For females, a significant regression equation was found (F_(2,1540)_ = 1519.4, *p* < 0.0001), with an R^2^ adjusted of 0.663, considering, at the end of the forward selection, only two variables. The variable associated with endomorphy values with a positive coefficient was BMI; the variable negatively associated was TBW%.

The female predicted endomorphy value measurement was:Endomorphy value = 4.313 − 0.0572 × TWB% + 0.145 × BMI

### 3.3. Mesomorph Somatotype

Multiple linear regression was calculated to predict mesomorph somatotype values. For males, a significant regression equation was found (F_(3,1278)_ = 1610.1, *p* < 0.0001), with an R^2^ adjusted of 0.790, considering, at the end of the forward selection, only three variables. The variable associated with mesomorphy values with a positive coefficient was BMI; the variables negatively associated were stature and Rz.

The male predicted mesomorphy value measurement was:Mesomorphy value = 11.81 − 0.0524 × Stature − 0.00725 × Rz + 0.230 × BMI

For females, a significant regression equation was found (F_(3,1538)_ = 3242.2, *p* < 0.0001), with an R^2^ adjusted of 0.863, considering, at the end of the forward selection, only three variables. The variable associated with endomorphy values with a positive coefficient was BMI; the variables negatively associated were stature and Rz.

The female predicted mesomorphy value measurement was:Mesomorphy value = 8.91 − 0.0589 × Stature − 0.00395 × Rz + 0.317 × BMI

### 3.4. Ectomorph Somatotype

Multiple linear regression was calculated to predict ectomorph somatotype values. For males, a significant regression equation was found (F_(6,1276)_ = 3967.4, *p* < 0.0001), with an R^2^ adjusted of 0.949, considering, at the end of the forward selection, six variables. The variables associated with ectomorphy values with a positive coefficient were stature, Rz, TBW%, hydration, and BMI; the variables negatively associated was TBW.

The male predicted ectomorphy value measurement was:Ectomorphy value = −60.25 + 0.188 × Stature + 0.0146 × Rz − 0.350 × TBW + 0.345 × TBW% + 0.4174 × BMI + 0.105 × Hydration

For females, a significant regression equation was found (F_(4,1538)_ = 1529.8, *p* < 0.0001), with an R^2^ adjusted of 0.802, considering, at the end of the forward selection, four variables. The variable associated with ectomorphy values with a positive coefficient were BMI, TWB%, and MM%; the variable negatively associated was FFMI.

The female predicted ectomorphy value measurement was:Ectomorphy value = −2.119 + 0.119 × TWB% + 0.0778 × MM% + 0.244 × BMI − 0.709 × FFMI

The multiple regression model predicted the endomorphy score with an R^2^ adjusted > 0.66. The goodness of fitting relative to the mesomorphy was higher than the one of the endomorphy (R^2^ adjusted > 0.79). Finally, the goodness of fitting of the ectomorphy was more significant than those of the mesomorphy and endomorphy (R^2^ adjusted > 0.80). The bioimpedance variables obtained through the BIVA were not very different from those obtained through the Heath–Carter method, confirming observations made previously in the general population and athletes [18,19]. Through a linear equation, these variables defined the endomorph, mesomorph, and ectomorph somatotypes. The degree of accuracy of the prediction was quantified using the somatochart, which is always used following the classic Heath–Carter method, to categorize the subjects according to the three different scores of the respective somatotypes (endomorphy, mesomorphy, and ectomorphy). Based on this method, each subject acquired a precise position on the graph. Figure 1 shows the somatochart obtained by skinfolds (left panel) and resulting from multiple linear regressions (right panel). Bivariate distribution in both charts were similar; a distortion was observed at the left-top area in the predicted graph. Even if this feature was undesirable, the predicted point correctly defined subjects with higher BMI. The overall goodness of fitting was better for males, while predicted values for females did not explain all the variability of the raw sample data.

Bland–Altman plots (Figure 2) showed agreement between the two methods reported above. X-predicted values were not biased, with a low amount of random error; Y-predicted, even if not biased, showed a higher random error. Error distribution was independent of abscissa values. For males, Lin’s concordance correlation coefficients (CCC) for x and y values of the somatocharts were 0.891 and 0.899, respectively; for females, the Lin’s CCC were 0.943 and 0.881, respectively.

In addition, Bland–Altman plots for each somatotype component (endomorphy, mesomorphy, and ectomorphy), stratified by sex, were calculated and reported as Appendix A.

## 4. Discussion

This study aimed to propose a new strategy for somatotype evaluation using a BIVA analysis in the Italian population (previous work has verified a similar correlation in the Russian population) [28] to reduce the potential limits related to the most used measurement techniques. The standard technique involves measuring the mass, stature, triceps, iliac crest and medial calf skinfolds, the girths of the arm (flexed and tensed) and calf, and the joint breadths of the biepicondylar humerus and biepicondylar femur. The correct execution of 11 measurements, manually detected, (obviously, every point should be repeated to minimize the standard error) represented a critical issue, due to the possibilities of errors (techniques involving the measurement of a single fold can lead to errors up to 150%; the evaluation of the girths showed a margin of error of 0.2 cm) [29]; the required time to do it was 30 to 40 min. BIVA has been validated against densitometry, with correlation coefficient values ranging from 0.907 to 0.952 and standard error of estimates from 1.97 to 3.03 kg [30]; more recently, assessments estimated the standard errors of the best BIA and BIVA regression equations to be ~3–6% for FFM and ~3–8% for TBW [17,26,31]. In addition to a good correlation with the reference techniques, the BIA and BIVA execution proved to be notably fast (around 3–5 min), and, considering the minimal intervention of the operator, able to reduce the potential errors related to the human factor significantly; it also limited the contact between the operator and the subject assessed, an essential factor, considering the recent pandemic scenario.

In the present study, the prediction of endomorph, mesomorph, and ectomorph somatotypes was obtained with significant regression equations. It suggests a possible automatic evaluation of the Heath–Carter somatotypes by BIA in adults.

Three variables were associated with endomorph in male subjects; in particular, BMI was positively correlated with this somatotype, while a negative correlation was found with stature and TBW%. In female subjects, only two variables were associated with endomorph: BMI was positively correlated, while a negative correlation was found only with TBW%. This somatotype has a significant relationship with the fat mass percentage, as reported in classical studies [32], and body fat mass can affect the BMI and TBW%, as reported in our data. Therefore, stature could most likely be related as a sex-related element in the definition of this somatotype. However, endomorphy values showed the lowest R^2^_adj_ values; this implies that the multiple regression model does not precisely explain the values obtained with anthropometric measurements.

A significant regression equation for the mesomorph somatotype has been found, considering the end of the best subset selection on three variables. The positive correlation of the mesomorph rating with BMI, an element not fully compatible, has also been found by Anisimova et al. [33]. The variables associated with mesomorph values with a negative coefficient were stature and Rz in male and female subjects. A significant relationship between the mesomorph component with lean body mass has been revealed, as reported by previous literature [33], defining a BMI–Rz balance that could be compatible with a subject with well-represented muscle masses.

The more significant regression equation was found for the ectomorph somatotype, considering, at the end of the best subset selection, six variables for males and four for females. In male subjects, stature, Rz, TBW%, hydration, and BMI were positively correlated with this somatotype, while a negative correlation was found with TBW. On the other hand, in female subjects, only four variables were associated with ectomorph: BMI, TWB%, and MM%, which were positively correlated. A negative correlation was found only with FFMI. In this case, the relationship created between stature and parameters directly or indirectly related to the levels of body fluids in the male and those related to the estimate of muscle mass in the female could explain the correlation between somatotype and electrical measurement with specific sex dynamics.

The sample size represented significant strengths, with examined data robustness, method repeatability, and the relative simplicity of the equations obtained, and above all, the parameters that correlated with the somatotype generally overlapped in the two genres, with a situation of greater complexity only in the ectomorph. The dispersion indicated by the graphs represents limits. The equations include various parameters derived from the resistance and reactance measurements, elements capable of reducing the accuracy of the evaluation.

Beyond the correlation of the somatotype, it was helpful to identify the location of the point on the graph calculated using the conventional Heath–Carter method and to compare it with the location of the point obtained from the BIA. According to the literature, the somatochart, both obtained by the Heath–Carter method and resulting from multiple linear regressions, was used; the latter can predict, with acceptable accuracy, the coordinates of each subject. Interestingly, the sample enrolled in this study did not follow any specific instructions (e.g., diet, hydration, physical activity) before the BIA measurements; this might increase the dispersion of the estimated parameters, but it represented a point of strength in this procedure, as the sample was more representative of the actual population. Furthermore, this method estimated the state of hydration, essential information which could not be known with standard and manual measurement techniques.

### Limitations

The execution of anthropometric measurements by a single operator constituted the limits of this study, as it remains to be verified whether a higher or lower level of overlap occurred by processing data collected by several operators; furthermore, the proposed method was applied when using bioimpedance devices operating at a single frequency of 50 kHz, using the detection protocol described. For future prospects, we suggest an evaluation of a larger sample, which would allow us to investigate any stratifications of data based on ethnicity, physical activity practiced, age, and health condition. In addition, it would be interesting to verify its applicability with other BIVA devices. These limitations were generalized for the bioimpedance method and are variable from instrument to instrument, impacting, on average, between 2 and 4%, in healthy subjects [34,35]. It remains important to consider how, in outpatient practice, it is essential to provide data, such as those treated in this work, that allow an evaluation of the general population and not on ultra-selected samples that can hardly find confirmation in daily clinical practice.

## 5. Conclusions

The results obtained with this analysis prove to be very interesting, as they suggest integrating the software dedicated to the BIA analysis with equations that allow us to derive, in addition to those already present, the somatotype. It provides a series of exciting advantages; these could be to the professional who uses techniques to significantly reduce detection timing, the possible margin of error due to manual skills, the different tools used, and finally, the contact with the subject to be evaluated. It also makes possible the evaluation in “self-analysis,” which is practically impossible today with the classic Heath–Carter method. The evaluation of the somatotype with this method could allow for obtaining more standardized data, suggesting that BIA analysis could be further investigated for future applications. Further studies will be needed to clarify and evaluate the actual usability of this and future new approaches to BIA analysis.

## Figures and Tables

**Figure 1 jfmk-07-00086-f001:**
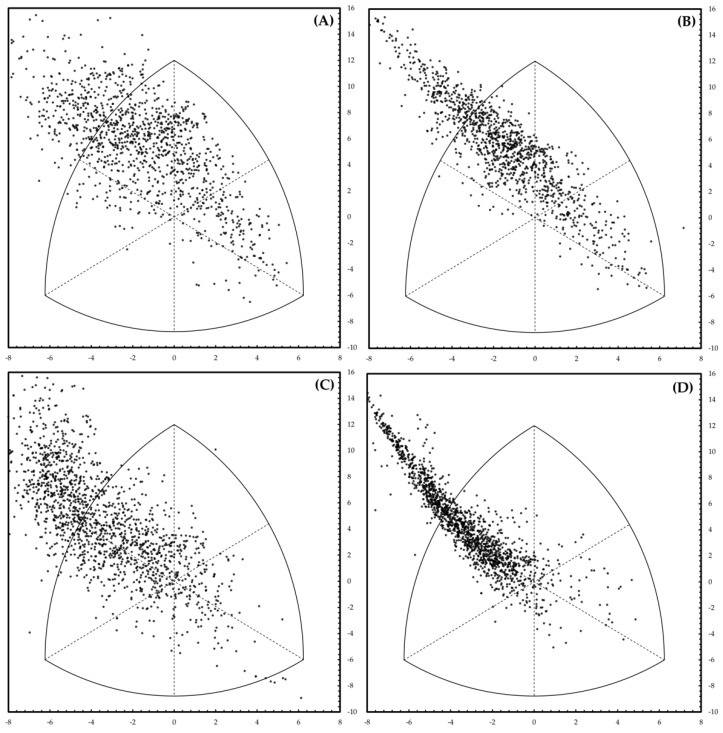
Somatocharts obtained by the Heath–Carter method (left panels) and resulting from multiple linear regressions (right panels) for male (above, panels (**A**,**B**)) and female (below, panels (**C**,**D**)) subjects.

**Figure 2 jfmk-07-00086-f002:**
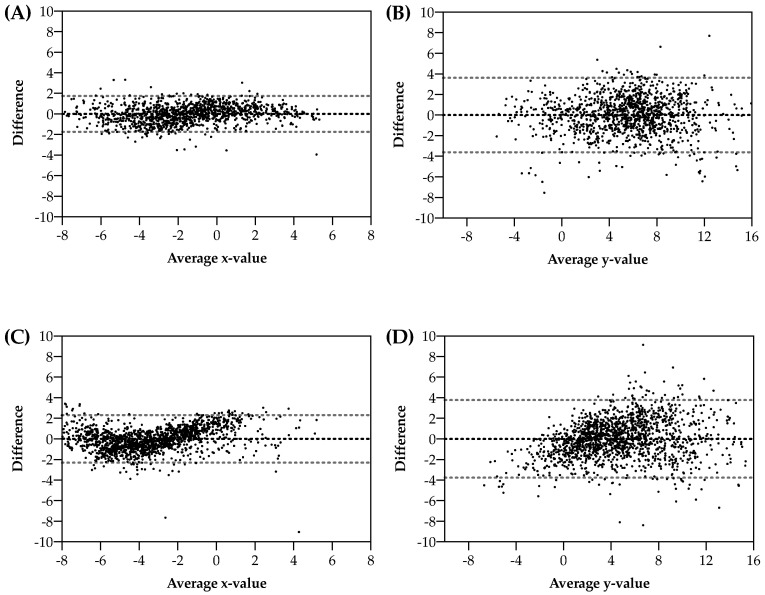
Agreement between (Bland–Altman plots) results obtained by the Heath–Carter method and resulting from multiple linear regressions for x- and y-axis, stratified for male (above, panels (**A**,**B**)) and female (below, panels (**C**,**D**)) subjects.

**Table 1 jfmk-07-00086-t001:** Participants’ demographic and anthropometric characteristics.

	Males (*n* = 1189)	Females (*n* = 1638)	*F*	*p* (*F*)	η^2^_p_
Age (years)	41.8 ± 15.8	43.5 ± 16.7	7.973	0.005	0.003
Stature (cm)	175.5 ± 7.9	162.2 ± 7.3	2046.509	<0.001	0.431
Body mass (kg)	79.2 ± 17.2	69.4 ± 16.6	334.905	<0.001	0.110
BMI (kg/m^2^)	26.4 ± 6.4	25.6 ± 5.3	2.454	0.117	0.001
Resistance (ohm)	420.3 ± 63.6	493.8 ± 68.2	816.334	<0.001	0.232
Reactance (ohm)	55.9 ± 11.7	57.1 ± 9.9	6.743	0.009	0.002
Phase angle (degrees)	7.6 ± 1.2	6.6 ± 1.0	570.559	<0.001	0.174
Free fat mass (kg)	67.0 ± 12.2	49.8 ± 6.0	2318.629	<0.001	0.461
Triceps skinfold (mm)	10.5 ± 4.6	18.8 ± 5.6	1720.156	<0.001	0.389
Subscapular skinfold (mm)	13.4 ± 6.1	14.7 ± 7.5	26.885	<0.001	0.010
Suprailiac skinfold (mm)	12.2 ± 5.9	13.6 ± 5.7	40.384	<0.001	0.015
Medial calf skinfold (mm)	7.1 ± 4.2	12.5 ± 4.5	1015.657	<0.001	0.273
Biepicondylar humerus width (cm)	5.6 ± 1.8	4.9 ± 0.4	168.076	<0.001	0.058
Arm circumference (cm)	30.5 ± 3.7	28.7 ± 4.1	123.038	<0.001	0.043
Biepicondylar femur width (cm)	10.0 ± 0.7	9.9 ± 2.4	2.937	0.087	0.001
Calf circumference (cm)	37.2 ± 3.2	36.6 ± 3.7	18.197	<0.001	0.007
Endomorphy	3.5 ± 1.4	4.8 ± 1.5	583.977	<0.001	0.177
Mesomorphy	5.5 ± 2.3	5.6 ± 2.5	1.690	0.194	0.001
Ectomorphy	1.9 ± 1.4	1.4 ± 1.3	90.950	<0.001	0.033

**Table 2 jfmk-07-00086-t002:** Prediction models for endomorph, mesomorph and ectomorph somatotype values.

Sex	Somatotype	Predictors	R^2^_adj_	SEE	VIF
Male	Endomorphy	Stature TWB% BMI	0.68	0.68	1.02 1.96 1.95
	Mesomorphy	Stature Rz BMI	0.79	0.57	1.01 1.41 1.40
	Ectomorphy	Stature Rz TBW TBW% BMI Hydration	0.95	0.09	1.17 6.59 4.83 9.17 12.50 5.60
Female	Endomorphy	TWB% BMI	0.66	0.74	2.69 2.78
	Mesomorphy	Stature Rz BMI	0.86	0.65	1.03 1.46 1.46
	Ectomorphy	TBW% MM% BMI FFMI	0.80	0.37	3.49 2.39 2.80 11.40

Abbreviations: TBW = total body water; BMI = body mass index; Rz = resistance; MM = muscular mass; FFMI = free fat mass index.

## Data Availability

Not applicable.

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
