# Peer review of "A New Strategy for Somatotype Assessment Using Bioimpedance Analysis: Stratification According to Sex"

_jfmk, 2022, doi:10.3390/jfmk7040086_

Round 1

Reviewer 1 Report

First of all, thank you to the authors for such an interesting study. I will now give you my review of the paper:
INTRODUCTION
- A definition of somatotype would be appreciated in the introduction. 
- The first paragraph of the introduction talks about body composition. However, somatotype does not only assess body composition but also body shape. Therefore, it is advisable to modify this first paragraph to point out the importance of somatotype in different domains.
- Throughout the introduction somatotype is used as a synonym for body composition. This should be modified with a thorough review of the literature.
- The introduction should discuss the different methods available for somatotype analysis, as well as their advantages and disadvantages.
- Why do you consider anthropometry as the gold-standard method for somatotype assessment? This might come as a surprise, so it should be justified in the introduction.
- The introduction is out of context of the study. It is advisable to restructure the information it contains.
METHOD:
- Could you justify the choice of the Lohman et al. anthropometric technique instead of more up-to-date, standardised and globally disseminated protocols such as the ISAK, which was used in Heath-Carter's own method? Why did you decide to use a different protocol than the original authors?
- Please adjust the name of the anthropometric variables to the correct international terminology.
- Please include photos of all anthropometric measurements included.
- The section concerning the taking of variables lacks important information such as: Who took the measurements? What training did they have? How many times did they do it? With what material? What was the final value taken for each variable? What was the intra- and inter-observer TEM? Were hydration, intake and time conditions standardised when performing the bioimpedance? If so, how was this done?
STATISTICAL ANALYSIS:
- Just as you have compared the somatopoint x and y values with the original method and with the prediction made by the equation, could you do the same for the individual components, including in the analysis as covariates those aspects that could be influencing the results such as BMI or sex? And do the Bland-Altman test for the individual components?
- Could you include a Lin's concordance correlation coefficient analysis both for the individual components and for the x and y coordinates of the somatotype?
RESULTS AND DISCUSSION
- In your opinion, are the R2 values high enough to support your assertions? I refer especially to the values for the linear regression of endomorphy.
- When you state "Figure 3 shows the somatochart obtained by skinfolds" this statement is not correct. The somatochart cannot be obtained by skinfolds alone. Adjust the terminology.
- I will continue to provide input to the discussion once the discussion has been modified with the requested analyses.

Author Response

Reviewer #1

First of all, thank you to the authors for such an interesting study. I will now give you my review of the paper:

INTRODUCTION

- A definition of somatotype would be appreciated in the introduction.

Answer: Thanks for this comment. The definition of somatotype was now provided (see lines 81-85).

- The first paragraph of the introduction talks about body composition. However, somatotype does not only assess body composition but also body shape. Therefore, it is advisable to modify this first paragraph to point out the importance of somatotype in different domains.

Answer: Thanks for this comment. Lines 41-43 have been added to a description that integrates morphological analysis with body composition with related application examples

- Throughout the introduction somatotype is used as a synonym for body composition. This should be modified with a thorough review of the literature.

Answer: Thanks for this comment, which allows us to make the manuscript clearer. We have integrated some sentences to make the differentiation between body composition and somatotype. It was due to an excessive simplification of the exposition, not a conceptual misunderstanding.

- The introduction should discuss the different methods available for somatotype analysis, as well as their advantages and disadvantages.

Answer: Thanks for this comment. Our intent is to evaluate whether, with a method that allows to minimize the intervention of an operator and reduces acquisition times and the possibility of error such as BIVA, it is possible to obtain results that can be superimposed with the method of Heath and Carter (based on anthropometric measurements) in the definition of the somatotype, not to make an evaluation of comparative techniques. The possible advantages that could derive from it are discussed later (lines 131-157 and 550-570). A comparative analysis of different techniques could, in our opinion, lead to an off-topic discussion.

- Why do you consider anthropometry as the gold-standard method for somatotype assessment? This might come as a surprise, so it should be justified in the introduction.

Answer: Thanks for this comment. In the original publications, anthropometry is the unique way provided by the method of Heath and Carter in the evaluation of the somatotype; our intent is to evaluate whether with the use of an outpatient technique such as BIVA it is possible to have comparable results in the reference population. We can probably have given this idea by saying that it is the most used method without specifying that this is applied on an outpatient basis; we have corrected this, and we apologize for the inattention.

- The introduction is out of context of the study. It is advisable to restructure the information it contains.

Answer: Thanks for this comment. The introduction talks about the somatotype technique based on anthropometric surveys and the bioimpedentiometry technique based on bioelectrical parameters that we used to evaluate whether it was possible to obtain comparable results in an outpatient setting in the reference population. Our intent is not to compare two techniques born to do the same thing but to study a possible new application potential for BIVA, as we added at the end of the Introduction. We thank the Reviewer as by applying the proposed corrections, the text is, in its intent, much clearer.

METHOD:

- Could you justify the choice of the Lohman et al. anthropometric technique instead of more up-to-date, standardised and globally disseminated protocols such as the ISAK, which was used in Heath-Carter's own method? Why did you decide to use a different protocol than the original authors?

Answer: We thank the reviewer for this comment, which allowed us to improve the quality of our work. The morphotype analysis according to Heath & Carter was defined in 1967 (ref 4), specifying the method to be used in the detection of anthropometric measurements, then reaffirmed by Carter’s manual in 2002 (ref 3), where the methods for detecting anthropometric measurements are reiterated to which we have followed. We have then decided to remain faithful to what was published by the original authors of the somatotype detection technique. The indications provided by Lohman do not substantially differ from the method proposed by the original authors and represent one of the guidelines closest to Carter's last publication in 2002. We appreciate the comment and agree on the inclusion of a more updated reference, so we have adhered to this: "Norton, Kevin. (2018). Standards for anthropometry assessment. Chapter 4 in: Kinanthropometry and Exercise Physiology, 4th ed, pp 68-137. 10.4324/9781315385662-4."

- Please adjust the name of the anthropometric variables to the correct international terminology.

Answer: We thank the reviewer for the report; we have replaced the terms used with those indicated in reference 23.

- Please include photos of all anthropometric measurements included.

Answer: We thank you for the report. Since it is the application of standardized techniques and described in the literature it was not considered necessary and therefore ethical to acquire photos, wanting to guarantee maximum privacy of the subjects evaluated. If the reviewer considers it essential, we can create some graphics relating to the measures acquired.

- The section concerning the taking of variables lacks important information such as: Who took the measurements? What training did they have? How many times did they do it? With what material? What was the final value taken for each variable? What was the intra- and inter-observer TEM? Were hydration, intake and time conditions standardised when performing the bioimpedance? If so, how was this done?

Answer: We thank you for the report, which allows us to improve the clarity and quality of the work. The required information has been added throughout the Methods section.

STATISTICAL ANALYSIS:

- Just as you have compared the somatopoint x and y values with the original method and with the prediction made by the equation, could you do the same for the individual components, including in the analysis as covariates those aspects that could be influencing the results such as BMI or sex? And do the Bland-Altman test for the individual components?

Answer: We agree with the reviewer’s comment. We have updated our results including six new Bland-Altman plots, related to each individual component (endomorph, mesomorph, ectomorph), stratified for sex, as requested; we added them as Supplementary Material. BMI was already considered as a covariate in the forward stepwise multiple regression model in all the elaboration performed.

- Could you include a Lin's concordance correlation coefficient analysis both for the individual components and for the x and y coordinates of the somatotype?

Answer: Thanks for this comment. We have calculated Lin’s coefficients, which are now reported in the results.

RESULTS AND DISCUSSION

- In your opinion, are the R2 values high enough to support your assertions? I refer especially to the values for the linear regression of endomorphy.

Answer: Thanks for this comment. We agree that R2 values for endomorphy were not extremely high; so, we have added the following sentence in the discussion section: “However, endomorphy values showed the lowest R2adj values; this implies that the multiple regression model does not precisely explain the values obtained with anthropometric measurements”.

- When you state "Figure 3 shows the somatochart obtained by skinfolds" this statement is not correct. The somatochart cannot be obtained by skinfolds alone. Adjust the terminology.

Answer: Thanks for the tip; we have replaced it with "the Heath-Carter method".

- I will continue to provide input to the discussion once the discussion has been modified with the requested analyses.

Reviewer 2 Report

Overall this is a well written paper with an interesting result anthropometric evaluation

The results are based on rational working hypothesis, well described and with a correct research design.

INTRODUCTION

The introduction provides sufficient background information for readers to understand the research aim, however the authors should clarify the relation between BC obtained by BIA and somatotype.

Motivations for this study are more than clear and the objectives are clearly defined at the Introduction, the argumentation in this part was concise.

METHODS

The methodology proposed to reach the aim of the study look appropriate, well designed and conducted.

Provide the Ethical Committee approval number of the study as well as the ethical principles followed.

Specify the manufactured, company and country of all the instrument used in the research.

There are a few instances where assertions are made that are not substantiated with references.

RESULTS

Results paragraph should include the most relevant data.

All of the tables include specific, good developed statistic.

DISCUSSION

All possible interpretations of the data considered are consistent, however there are some grammar mistakes that should be corrected.

Conclusion should respond the research aim

Explain limitation of the study and future research line according to the study conclusion

LITERATURE CITED

The literature cited is relevant to the study, but there are several instances in which the author makes assertions without substantiating them with references, but which are sustained by the main text and previous citations.

Reference style should be checked with the journal standards. 

Author Response

Reviewer #2

Overall this is a well written paper with an interesting result anthropometric evaluation

The results are based on rational working hypothesis, well described and with a correct research design.

INTRODUCTION

The introduction provides sufficient background information for readers to understand the research aim, however the authors should clarify the relation between BC obtained by BIA and somatotype.

Answer: Thanks for this comment. The introduction has been modified according to this and other reviewers’ comments.

Motivations for this study are more than clear and the objectives are clearly defined at the Introduction, the argumentation in this part was concise.

METHODS

The methodology proposed to reach the aim of the study look appropriate, well designed and conducted.

Provide the Ethical Committee approval number of the study as well as the ethical principles followed.

Answer: Thanks for this comment. The approval number by the Ethics Committee is reported in Section 2.1, as well as the Helsinki declaration. These statements are also reported at the end of the manuscript, in the section “Institutional Review Board Statement”.

Specify the manufactured, company and country of all the instrument used in the research.

Answer: We thank you for your comment, which allows us to improve the clarity and quality of the work. The required information has been added to section 2.2.

There are a few instances where assertions are made that are not substantiated with references.

Answer: We thank you for the report. We have tried to integrate them.

RESULTS

Results paragraph should include the most relevant data. All of the tables include specific, good developed statistic.

Answer: We thank you for these comments.

DISCUSSION

All possible interpretations of the data considered are consistent, however there are some grammar mistakes that should be corrected.

Answer: Thanks for this comment. We have revised the manuscript to correct the errors.

Conclusion should respond the research aim

Answer: Thanks for this comment. The conclusion was adjusted accordingly to this comment.

Explain limitation of the study and future research line according to the study conclusion

Answer: Thanks for this comment. A Limitations section has been added at the end of the Discussion.

LITERATURE CITED

The literature cited is relevant to the study, but there are several instances in which the author makes assertions without substantiating them with references, but which are sustained by the main text and previous citations.

Answer: We thank you for the report that allowed us to improve the quality of the work. We have set the text in this way to avoid being redundant with quotes already made; however, we have intervened in the introductory part and in that of materials and methods by inserting the references in support of the statements essential for the purposes of the processing.

Reference style should be checked with the journal standards.

Answer: The references were checked and formatted according to the journal guidelines.

Reviewer 3 Report

STRUCTURE

-       The manuscript is properly structured.

TITLE AND ABSTRACT

- The title or abstract should inform that the type of study.

- The word "sex" refers to the biological and physiological characteristics that define men and women, while "gender" refers to the socially constructed roles, behaviours, activities and attributes that a given culture considers appropriate for men and women. It is recommended that "gender" be changed to "sex".

INTRODUCTION

-       Line 76: the introduction discusses the advantages of bioelectrical impedance (BIVA), however, it does not specify its disadvantages or drawbacks, e.g. in this technique it is assumed:

o   That the body is a cylinder.

o   Biological tissues act as conductors.

In addition, there are factors or requirements that can modify the body water content and distribution of body water:

o   Not ingesting food and drink 4 h before the test.

o   Urinate 30 min before the test.

o   Not being dehydrated.

o   No exercise in the previous 12 hours.

o   Assess the phase of the woman's menstrual cycle.

o   Check the ambient temperature is around 25ºC.

o   Observe that arms and legs are apart.

o   The subject should be isolated from metallic objects.

o   The individual's skin should be cleaned with alcohol.

-       There is no mention of differences in body composition between women and men, and how this variable might affect the results.

-       The literature search is brief and most of them are more than 5 years old. Updating the bibliography

o   Campa F, Matias CN, Nikolaidis PT, Lukaski H, Talluri J, Toselli S. Prediction of Somatotype from Bioimpedance Analysis in Elite Youth Soccer Players. Int J Environ Res Public Health. 2020 Nov 5;17(21):8176. doi: 10.3390/ijerph17218176. PMID: 33167449; PMCID: PMC7663908.

o   Rudnev SG, Negasheva MA, & Godina EZ. Assessment of the Heath-Carter somatotype in adults using bioelectrical impedance analysis. In Journal of Physics: Conference Series. 2019 July 1 (1272):012001). doi:10.1088/1742-6596/1272/1/012001.

MATERIALS AND METHODS

2.1. Participants

-       It is recommended to add table 1 under "Results", as it shows the characteristics of the participants.

Inclusion criteria

-       The article does not show what the inclusion or exclusion criteria were for the research.

2.2. Anthropometric assessment.

- What was the precision used? For example, “height was measured to the nearest 0.1 cm using an anthropometer (specify the material used)”.

- What is the anthropometrist's level? The entire methodology of the study should be stated in such a way that it can be replicated. For example, “All measurements were performed by the same investigator, an ISAK level 2 anthropometrist. The mean technical error was less than 1% for perimeters, circumferences, lengths, and heights and less than 5% for skinfolds.”

RESULTS

-       Line 168: 3.1. MulticollinearityIt. This paragraph mentions the following variable: nutrition. However, it has not been discussed above in the methods section.

-       Table 1. Perhaps one could start the paragraph as follows: A total of 1189 male and 1638 female participants took part in this study: 42,1% male and 57,9% female. Table 1 shows the basic anthropometric measurements. The mean weight is 79,2 ± 17,2 kg and 69,4 ± 16,6 kg for males and females, respectively. Use the same procedure for height. Briefly comment on the results. For example, males show higher values, presenting significant differences (if significant and add p-value). The following statistical analyses could be performed: Cohen's d (effect size); mean differences were significant at p < 0.05 and t student. In table 1 you can also add the anthropometric values.

-       It is recommended to add a second table with the prediction models for endomorphy, mesomorphy and ectomorphy based on anthropometric and bioimpedance variables, including the variance inflation factor, multiple regression model, and the rest of the analyses and variables used. This article can serve as an example:

o   Campa F, Matias CN, Nikolaidis PT, Lukaski H, Talluri J, Toselli S. Prediction of Somatotype from Bioimpedance Analysis in Elite Youth Soccer Players. Int J Environ Res Public Health. 2020 Nov 5;17(21):8176. doi: 10.3390/ijerph17218176. PMID: 33167449; PMCID: PMC7663908.

-       Comment on the results obtained from Figure 1.

DISCUSSION

-       Provide a cautious overall interpretation of the results taking into account the objectives, multiplicity of analyses, results of similar studies, and other relevant evidence. Add more information from other studies and researchers in the area and compare them with your results.

-       Line 299: “we found it helpful and exciting to identify the location of the point on the graph calculated…” Do not use the first person plural.

Limitations

-       The study does not point out its limitations, e.g. BIA was performed using a single frequency and does not guarantee reproducibility of results with different impedance devices.

CONCLUSIONS

-       The conclusions are from the research carried out, so it is not appropriate to cite other studies in this paragraph.

REFERENCES

-       References follow the indicated style

Author Response

Reviewer #3

STRUCTURE

-       The manuscript is properly structured.

TITLE AND ABSTRACT

- The title or abstract should inform that the type of study.

Answer: Thanks for this comment. The type of study (i.e., observational) was added in the abstract.

- The word "sex" refers to the biological and physiological characteristics that define men and women, while "gender" refers to the socially constructed roles, behaviours, activities and attributes that a given culture considers appropriate for men and women. It is recommended that "gender" be changed to "sex".

Answer: Thanks for this comment. We agree with the reviewer, then we have replaced the term “gender” with “sex” throughout the manuscript.

INTRODUCTION

- Line 76: the introduction discusses the advantages of bioelectrical impedance (BIVA), however, it does not specify its disadvantages or drawbacks, e.g. in this technique it is assumed:

o That the body is a cylinder.

o Biological tissues act as conductors.

In addition, there are factors or requirements that can modify the body water content and distribution of body water:

o   Not ingesting food and drink 4 h before the test.

o   Urinate 30 min before the test.

o   Not being dehydrated.

o   No exercise in the previous 12 hours.

o   Assess the phase of the woman's menstrual cycle.

o   Check the ambient temperature is around 25ºC.

o   Observe that arms and legs are apart.

o   The subject should be isolated from metallic objects.

o   The individual's skin should be cleaned with alcohol.

Answer: We thank you for the report. The requested information has been added to lines 131-150 and in the methods section, where relevant.

-       There is no mention of differences in body composition between women and men, and how this variable might affect the results.

Answer: We thank you for the comment. The requested information has been added to lines 164-168.

-       The literature search is brief and most of them are more than 5 years old. Updating the bibliography

o   Campa F, Matias CN, Nikolaidis PT, Lukaski H, Talluri J, Toselli S. Prediction of Somatotype from Bioimpedance Analysis in Elite Youth Soccer Players. Int J Environ Res Public Health. 2020 Nov 5;17(21):8176. doi: 10.3390/ijerph17218176. PMID: 33167449; PMCID: PMC7663908.

o   Rudnev SG, Negasheva MA, & Godina EZ. Assessment of the Heath-Carter somatotype in adults using bioelectrical impedance analysis. In Journal of Physics: Conference Series. 2019 July 1 (1272):012001). doi:10.1088/1742-6596/1272/1/012001.

Answer: We thank you for the comment. The suggested references have been added.

MATERIALS AND METHODS

2.1. Participants

- It is recommended to add table 1 under "Results", as it shows the characteristics of the participants.

Answer: Thanks for this comment. As suggested, we moved Table 1 under the Results section.

Inclusion criteria

- The article does not show what the inclusion or exclusion criteria were for the research.

Answer: Thanks for this comment. The inclusion and exclusion criteria have been reported.

2.2. Anthropometric assessment.

- What was the precision used? For example, “height was measured to the nearest 0.1 cm using an anthropometer (specify the material used)”.

Answer: Thanks for this comment. The requested information has been reported in the text.

- What is the anthropometrist's level? The entire methodology of the study should be stated in such a way that it can be replicated. For example, “All measurements were performed by the same investigator, an ISAK level 2 anthropometrist. The mean technical error was less than 1% for perimeters, circumferences, lengths, and heights and less than 5% for skinfolds.”

Answer: We thank you for the report, which allows us to improve the clarity and quality of the work. The required information has been added to the text.

RESULTS

-       Line 168: 3.1. Multicollinearity. This paragraph mentions the following variable: nutrition. However, it has not been discussed above in the methods section.

Answer: “Nutrition” value, as reported in the multicollinearity and in the statistical analyses sections refers to a parameter automatically provided by the bioimpedance analysis. So, we did not find it necessary to explain it further, also considering that it did not enter into any of the models.

-       Table 1. Perhaps one could start the paragraph as follows: A total of 1189 male and 1638 female participants took part in this study: 42,1% male and 57,9% female. Table 1 shows the basic anthropometric measurements. The mean weight is 79,2 ± 17,2 kg and 69,4 ± 16,6 kg for males and females, respectively. Use the same procedure for height. Briefly comment on the results. For example, males show higher values, presenting significant differences (if significant and add p-value). The following statistical analyses could be performed: Cohen's d (effect size); mean differences were significant at p < 0.05 and t student. In table 1 you can also add the anthropometric values.

Answer: Thanks for this comment. As suggested, we have added a sentence at the beginning of the Results section, before Table 1. Statistics have been calculated, and t-values, p-values, and effect sizes were reported.

-       It is recommended to add a second table with the prediction models for endomorphy, mesomorphy and ectomorphy based on anthropometric and bioimpedance variables, including the variance inflation factor, multiple regression model, and the rest of the analyses and variables used. This article can serve as an example:

o   Campa F, Matias CN, Nikolaidis PT, Lukaski H, Talluri J, Toselli S. Prediction of Somatotype from Bioimpedance Analysis in Elite Youth Soccer Players. Int J Environ Res Public Health. 2020 Nov 5;17(21):8176. doi: 10.3390/ijerph17218176. PMID: 33167449; PMCID: PMC7663908.

Answer: Thanks for this comment. We have added the Table with the model’s values, as suggested.

-       Comment on the results obtained from Figure 1.

Answer: Thanks for this comment. A brief comment about Figure 1 has been added, as requested.

DISCUSSION

-       Provide a cautious overall interpretation of the results taking into account the objectives, multiplicity of analyses, results of similar studies, and other relevant evidence. Add more information from other studies and researchers in the area and compare them with your results.

Answer: Since previous papers had low sample sizes, it’s not easy to compare this with our research; furthermore, the sample used was often belonging to a specific population (e.g., different sports players), so the equations obtained were not applicable to a broad population, and this makes its comparison with our results even more difficult. Some of the previously published papers in this field are the following:

Bolonchuk WW, Hall CB, Lukaski HC, Siders WA. Relationship between body composition and the components of somatotype. Am J Hum Biol. 1989;1(3):239-248. doi: 10.1002/ajhb.1310010303. PMID: 28514097.

Campa F, Bongiovanni T, Matias CN, Genovesi F, Trecroci A, Rossi A, Iaia FM, Alberti G, Pasta G, Toselli S. A New Strategy to Integrate Heath-Carter Somatotype Assessment with Bioelectrical Impedance Analysis in Elite Soccer Player. Sports (Basel). 2020 Oct 27;8(11):142. doi: 10.3390/sports8110142. PMID: 33121135; PMCID: PMC7694105.

Campa F, Matias CN, Nikolaidis PT, Lukaski H, Talluri J, Toselli S. Prediction of Somatotype from Bioimpedance Analysis in Elite Youth Soccer Players. Int J Environ Res Public Health. 2020 Nov 5;17(21):8176. doi: 10.3390/ijerph17218176. PMID: 33167449; PMCID: PMC7663908.

Castañeda Babarro A, Viribay Morales A, León Guereño P, Mielgo-Ayuso J, Urdampilleta A, Coca Núñez A. Anthropometric profile, body composition, and somatotype in stand-up paddle (SUP) boarding international athletes: a cross-sectional study. Nutr Hosp. 2020 Oct 21;37(5):958-963. English. doi: 10.20960/nh.03021. PMID: 32960636.

Kim CH, Park JH, Kim H, Chung S, Park SH. Modeling the human body shape in bioimpedance vector measurements. Annu Int Conf IEEE Eng Med Biol Soc. 2010;2010:3872-4. doi: 10.1109/IEMBS.2010.5627664. PMID: 21097071.

Urdampilleta A, Mielgo-Ayuso J, Valtueña J, Holway F, Cordova A. BODY COMPOSITION AND SOMATOTYPE OF PROFESSIONAL AND U23 HAND BASQUE PELOTA PLAYERS. Nutr Hosp. 2015 Nov 1;32(5):2208-15. doi: 10.3305/nh.2015.32.5.9602. PMID: 26545679.

-       Line 299: “we found it helpful and exciting to identify the location of the point on the graph calculated…” Do not use the first person plural.

Answer: Thanks for this comment. We have changed the sentence accordingly.

Limitations

-       The study does not point out its limitations, e.g. BIA was performed using a single frequency and does not guarantee reproducibility of results with different impedance devices.

Answer: Thanks for this comment. A section on Limitations has been added at the end of the Discussion.

CONCLUSIONS

-       The conclusions are from the research carried out, so it is not appropriate to cite other studies in this paragraph.

Answer: Thanks for the comment. The reference has been deleted.

REFERENCES

-       References follow the indicated style

Round 2

Reviewer 1 Report

The paper could be accepted in the current form

Reviewer 3 Report

No further comments.